# Basic fibroblast growth factor helps protect facial nerve cells in a freeze-induced paralysis model

Shinji Iwata[1], Hiroyuki Yamada[1]*, Masato Teraoka[1], Takemichi Tanaka[1], Takuya Kimura[1], Tomonori Joko[1], Yasuhiko Tabata[2], Hiroyuki Wakisaka[3], Naohito Hato[1]

1 Department of Otolaryngology, Head and Neck Surgery, Ehime University Graduate School of Medicine, Toon, Japan, 2 Laboratory of Biomaterials, Department of Regeneration Science and Engineering Institute for Frontier Life and Medical Sciences, Kyoto University, Kyoto, Japan, 3 Laboratory of Head and Neck Surgery, Ehime Prefectural University of Health Sciences, Ehime, Japan

* yamada.hiroyuki.mk@ehime-u.ac.jp

**Data Availability Statement:** The data underlying the results presented in the study are available from Department of Otolaryngology, Head and Neck Surgery, Ehime University Graduate School

## Abstract

Severe axonal damage in the peripheral nerves results in retrograde degeneration towards the central side, leading to neuronal cell death, eventually resulting in incomplete axonal regeneration and functional recovery. Therefore, it is necessary to evaluate the facial nerve nucleus in models of facial paralysis, and investigate the efficacy of treatments, to identify treatment options for severe paralysis. Consequently, we aimed to examine the percentage of facial nerve cell reduction and the extent to which intratympanic administration of a basic fibroblast growth factor (bFGF) inhibits neuronal cell death in a model of severe facial paralysis. A severe facial paralysis model was induced in Hartley guinea pigs by freezing the facial canal. Animals were divided into two groups: one group was treated with gelatin hydrogel impregnated with bFGF (bFGF group) and the other was treated with gelatin hydrogel impregnated with saline (control group). Facial movement scoring, electrophysiological testing, and histological assessment of facial neurons were performed. The freezing-induced facial paralysis model showed a facial neuronal cell death rate of 29.0%; however, bFGF administration reduced neuronal cell death to 15.8%. Facial movement scores improved in the bFGF group compared with those in the control group. Intratympanic bFGF administration has a protective effect on facial neurons in a model of severe facial paralysis. These findings suggest a potential therapeutic approach for treating patients with refractory facial paralysis. Further studies are required to explore the clinical applicability of this treatment.

## Introduction

Facial paralysis results in the loss of facial expressions, which affects a patient's quality of life. Some cases of facial paralysis are peripheral, and the main lesion is often located within the temporal bone [1]. In such cases, the pathology progresses within a narrow bony nerve canal. Inflammation of the facial nerve induces swelling within the nerve canal, leading to nerve constriction and ischemia, which exacerbates the paralysis [2]. Standard treatment for facial

of Medicine(Shinji Iwata, E-mail:h447033x@mails.
cc.ehime-u.ac.jp.

**Funding:** The author(s) received no specific
funding for this work.

**Competing interests:** The authors have declared
that no competing interests exist.

paralysis is the administration of glucocorticoids and antivirals; however, their effectiveness is limited in severe cases. Therefore, treatment of severe palsy is crucial to improve the rate of complete recovery from facial paralysis.

A previous study intratympanically administered a basic fibroblast growth factor (bFGF)-impregnated gelatin hydrogel within the temporal bone in a guinea pig model of facial paralysis and reported an accelerated improvement in facial paralysis and healing of the damaged area [3]. Another clinical study observed an enhanced complete healing rate in patients with severe Bell's palsy who received bFGF-impregnated gelatin hydrogel implants compared to those who underwent facial nerve decompression surgery alone [4]. These findings imply that intratympanic administration of bFGF for facial paralysis may improve palsy and facilitate healing of the damaged areas, indicating its therapeutic promise.

When axonal damage occurs in the peripheral nerves, Wallerian degeneration occurs on the peripheral side. However, depending on the extent of damage, retrograde degeneration occurs towards the central side, leading to neuronal cell death [5–7]. When this process is significant, axonal regeneration and functional recovery remain incomplete. Moreover, bFGF has various effects on neuronal cells, including axon elongation, induction of stem cell differentiation, and neuronal protection [8, 9].

However, to date, no studies have evaluated the facial nerve nucleus in models of facial paralysis involving bFGF administration. Therefore, it is necessary to evaluate the facial nerve nucleus in models of facial paralysis and investigate the efficacy of treatments, as this may increase treatment options in cases of severe paralysis.

Consequently, in this study, we aimed to examine the percentage reduction of facial nerve cells and the extent to which intratympanic administration of bFGF inhibits neuronal cell death in a model of severe facial paralysis.

## Materials and methods

### Animals

Female Hartley guinea pigs (n = 42), aged 8–10 weeks and weighing 400–500 g, were obtained from Japan SLC (Shizuoka, Japan). The animals were allowed access to standard laboratory food and water with an artificial 12-hour light/dark cycle before and after the experiments. The 42 guinea pigs were randomly divided into the saline-treated (control) and bFGF-treated groups.

All animal experiments were conducted in accordance with institutional requirements and ARRIVE 2.0 guidelines for animal research reporting [10]. The study protocol was approved by the Animal Care and Use Committee of the Ehime University School of Medicine (No. 05HI78-4).

### Surgical procedures

Facial paralysis was induced as previously described [11]. Each animal was anesthetized with an intramuscular injection of ketamine hydrochloride (35 mg/kg) and xylazine hydrochloride (7 mg/kg). The absence of the laryngeal reflex was used to indicate sufficient anesthesia, minimizing animal distress, before starting the procedure. Surgery was performed only on the left side, while the right side remained untouched.

The temporal bulla of the left ear was opened via a retro auricular approach, the freezing location was normalized, and the distance between the tip of the nozzle and facial canal was set to 5 mm. The vertical segment of the facial canal was then frozen for 5 s using a freeze spray (Oken, Fukuoka, Japan) without removing any bony wall. The freezing location was normalized, and the distance between the tip of the nozzle and facial canal was set to 5 mm. A gelatin

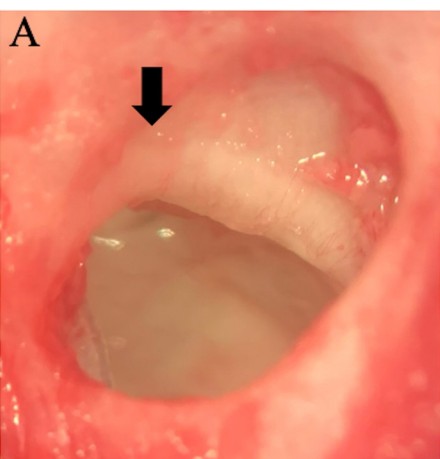
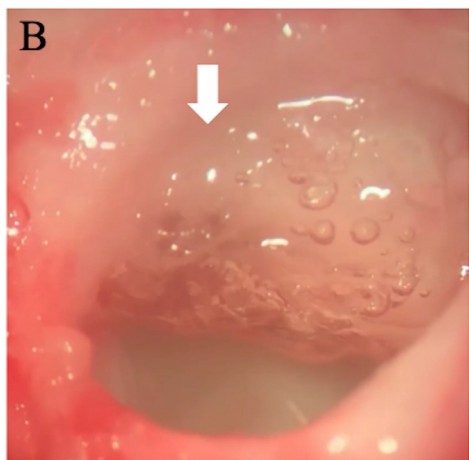

**Fig 1.** (a) Facial nerve canal (black arrow); (b) gelatin hydrogel with bFGF implanted over the facial nerve canal (white arrow).

hydrogel (sheet measuring 5 mm×5 mm) impregnated with 125 μg/125 μl bFGF (Fiblast Spray®; Kaken Pharmaceutical, Tokyo, Japan) was applied to the facial canal in the bFGF-treated group. The bFGF concentration was determined based on a previous study [3]. This gelatin hydrogel gradually released bFGF over approximately 2 weeks [12]. Conversely, a gelatin hydrogel impregnated with 125 μl of saline was applied to the facial canal in the saline-treated group (control group). The wound was closed with sutures (Fig 1). The animals that underwent treatment were divided into groups based on the amount of time until their histological assessment, conducted at the same intervals in the treatment and control groups, as follows: bFGF groups post-1 week (n = 7), post-4 weeks (n = 7), and post-10 weeks (n = 7); control groups post-1 week group (n = 7), post-4 weeks (n = 7), and post-10 weeks (n = 7).

## Facial movement evaluation

Facial movement was evaluated in both groups at 10 weeks (n = 7 per group). Facial paralysis severity was graded using the movements of the facial mimetic muscles at three sites (eyelid, lip, and nose) according to a previously described grading system [3]. Eyelid movement was evaluated using the blink response to air blown onto the eye using a 5-ml syringe. The degree of blink response was scored on a scale of 0–2 (0, blink reflex absent; 1, weak or delayed blink reflex; and 2, no between-side difference in blink reflex). Nose and lip movements were observed for 30 s each and scored on a scale of 0–2 (0: no nose or lip movements; 1: weaker movements than those on the healthy side; 2: no between-side difference in movements). The total facial movement score was defined as the sum of the eyelid-, nose-, and lip-movement scores. This measurement was performed in all experimental animals by one assessor, who had not been blinded to group allocation, and recorded weekly during the postoperative period.

## Electrophysiological testing

Electroneurography (ENoG) values were used to predict the severity of nerve palsy and time to recovery. They were calculated by measuring the amplitude of the compound motor action potential (CMAP) obtained by electrical stimulation of the left and right facial nerve trunks. All experimental animals were anesthetized with an intramuscular injection of ketamine hydrochloride (35 mg/kg) and ENoG values were recorded. The CMAP of the facial nerve was

recorded via the orbicularis oris muscle for each animal in both groups at 1 week postoperatively using electromyography (Panlab Le12106 Digital Stimulator, Harvard Apparatus, Holliston, Massachusetts, United States). The recording electrode was inserted into the orbicularis oris muscle, and the ground electrode was placed in the latissimus dorsi muscle. The facial nerve was stimulated at the mastoid foramen with rectangular current pulses at 0.1 ms, and EMG of the orbicularis oculi muscle was recorded with a sampling rate of 50 kHz and amplification range from ± 200 µV to ± 20 mV. The ENoG value was calculated using the following formula: 100×CMAP (affected side)/CMAP (healthy side).

## Histological assessment

The animals were transcardially perfused with saline followed by 4% paraformaldehyde in 0.1 M phosphate buffer (pH 7.4) under deep anesthesia. The brainstem tissue containing the facial nerve nucleus was dissected from the skull and fixed in the same solution overnight (4˚C). The brainstem tissue was then embedded in paraffin and serially sectioned at 10-µm-thick coronal planes using a microtome (Leica SM 2010R, Leica Biosystems, Japan). Every sixth section through the facial nucleus was stained with cresyl violet and analyzed to count the number of facial neurons using ImageJ (U. S. National Institutes of Health, Bethesda, Maryland, USA). Facial nerve cell counts were performed as previously described [13, 14]. Cells < 20 µm in diameter were excluded, and only those with clearly stained nuclei were counted to avoid counting cells other than facial neurons (Fig 2). In addition, the facial neuron mortality rate for each group was calculated according to the following formula: Facial neuron mortality rate = 1 −(Number of facial neurons in left side/Number of facial neurons in right side) × 100%.

## Statistical analysis

All statistical analyses were performed using EZR (Saitama Medical Center, Jichi Medical University, Saitama, Japan), which is a graphical user interface for R (The R Foundation for Statistical Computing, Vienna, Austria). Specifically, it is a modified version of the R commander, designed to add statistical functions that are frequently used in biostatistics. Intergroup comparison of facial movement evaluation was performed using the Mann–Whitney U test. A two-way analysis of variance was employed for intergroup comparisons of the facial neuron

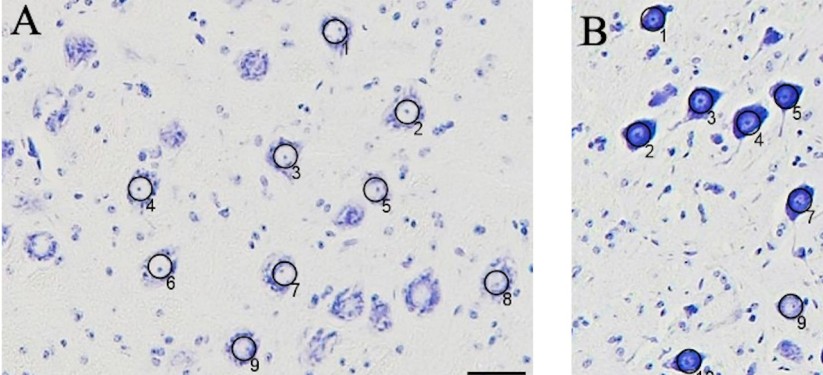

**Fig 2. Nerve cells in the facial nerve nucleus of the control model at 1 week.** (a) On the healthy side, each nerve cell exhibited a distinct white nucleus, internal nucleoli, and Nissl bodies in the cytoplasm. (b) On the affected side, numerous cells with densely stained cytoplasm were observed. Cells with nuclei and nucleoli that stained clearly were counted, and those with a diameter smaller than 20 µm were excluded. The circles and numbers represent the counted cells (underbar 50 µm).

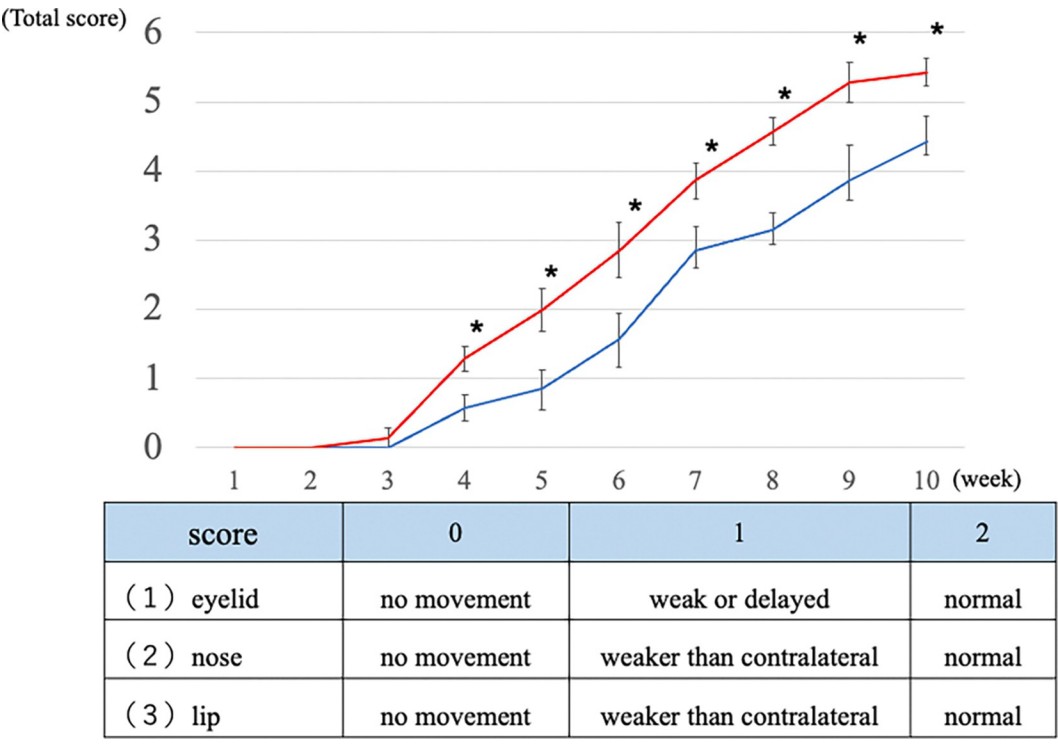

**Fig 3. Facial movement evaluation.** The red and blue lines represent the bFGF and control groups, respectively. The results of the bFGF group were better than those of the control group after 4 weeks (asterisk: p < 0.05).

mortality rate. In addition, a one-way analysis of variance (Bonferroni method) was used to test for significant differences among the groups. Statistical significance was set at p < 0.05.

## Results

### Facial movement evaluation and electrophysiological testing

Fig 3 summarizes the facial movement scores of each group at 10 weeks. Facial canal freezing caused complete facial paralysis in all animals at postoperative week 1. The facial movement scores in both groups increased from week 3–4. At 10 weeks, the score for the bFGF group (mean ± standard deviation, 5.4 ± 0.2 points) was significantly higher than that of the control group (4.4 ± 0.4 points; p < 0.05). Fig 4 shows the CMAP recordings of the healthy (right) and affected (left) sides. At postoperative 1 week, the ENoG value was 0% in both groups.

### Histological assessment

Fig 5 shows the nerve cells in the facial nerve nucleus of the control and bFGF groups at 1, 4, and 10 weeks. Each nerve cell exhibited distinct white nuclei (black arrow), internal nucleoli (red arrow), and Nissl bodies in the cytoplasm (blue arrow) on the healthy side. Meanwhile, numerous cells with densely stained cytoplasm (black arrowhead) were observed on the affected side; in addition, several cells had irregular shapes.

The cell reduction rates at 1 week were 3.8 ± 4.7% and 2.5 ± 1.9% for the control and bFGF groups, respectively, which were comparable. However, at 4 and 10 weeks, the corresponding values were 21.1 ± 4.8% and 13.9 ± 2.5% and 29.0 ± 2.6% and 15.8 ± 2.0% for the control and bFGF groups, respectively. The differences between the groups at 4 and 10 weeks were

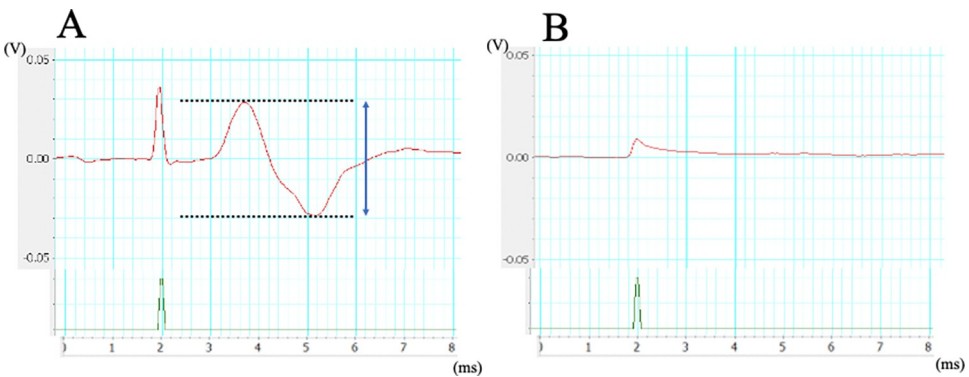

**Fig 4.** Recorded waveform: (a) healthy side and (b) affected side at postoperative 1 week. The arrow from the dotted line to the dotted line represents the amplitude. The ENoG value was 0% in the control and bFGF groups. The example shown is from the control group.

statistically significant. Moreover, within the control group, there was a significant difference in the facial neuron mortality rate between the 4- and 10-week assessments, whereas no such difference was observed in the bFGF group (Fig 6).

## Discussion

bFGF has varied functions, including supporting nerve cell survival and growth [8, 9]. Therefore, in this study, we investigated the effects of bFGF administration into the tympanic cavity on suppression of neuronal cell death in the facial nucleus. Our results showed a 29.0% reduction in the number of facial nerve cells in a freezing-induced facial paralysis model, which was mitigated to 15.8% by bFGF administration.

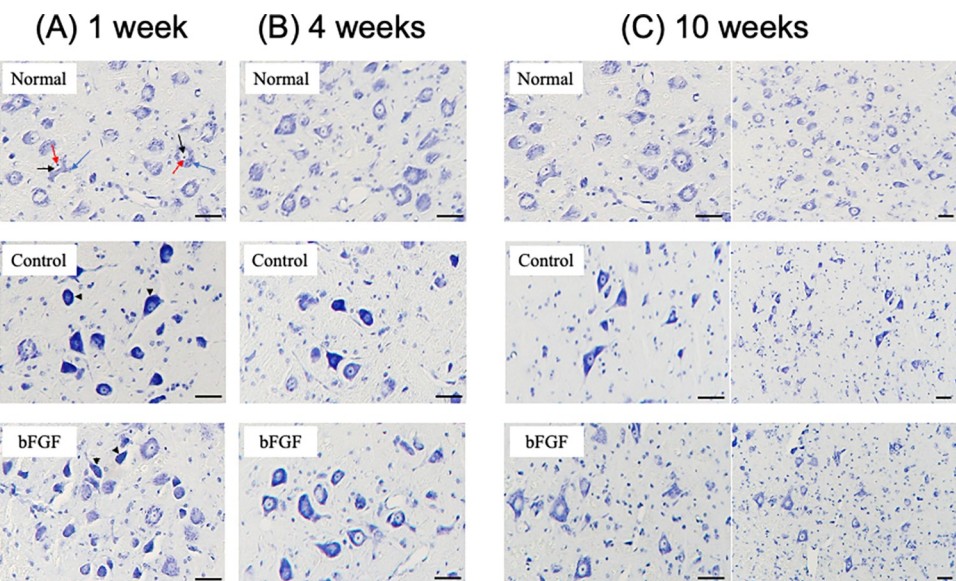

**Fig 5.** Representative images of the histological sections at (a) 1 week, (b) 4 weeks, and (c) 10 weeks postoperatively. Each nerve cell exhibited distinct white nuclei (black arrow), internal nucleoli (red arrow), and Nissl bodies in the cytoplasm (blue arrow) on the healthy side. In contrast, numerous cells with densely stained cytoplasm (black dotted arrow) were observed on the affected side. "Normal" means the healthy side of the control group. The number of facial nerve cells on the affected side in both the control and bFGF groups decreased at 4 weeks and 10 weeks postoperatively, and several cells had irregular shapes at 10 weeks (underbar 50 μm).

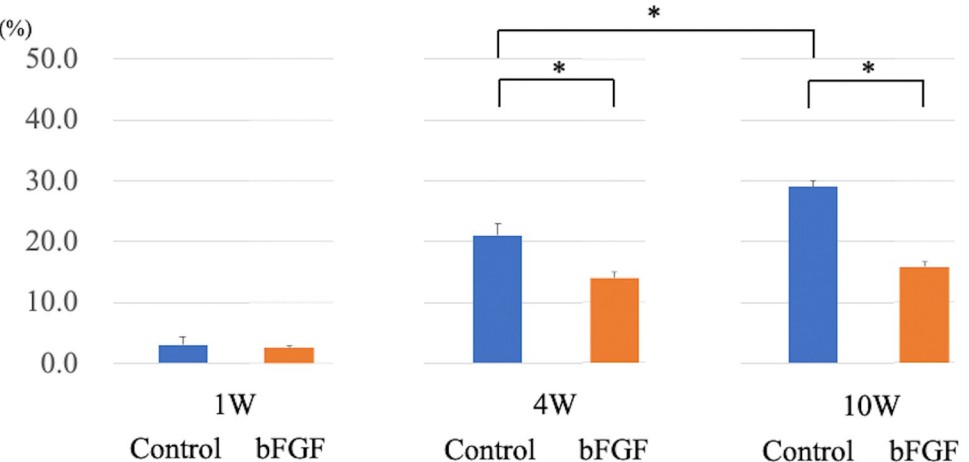

**Fig 6. Facial neuron mortality rate.** The cell reduction rates were as follows: at 1 week, the control group had a rate of 3.8 ± 4.7%, while the bFGF group had a rate of 2.5 ± 1.9%. At 4 weeks, the corresponding values were 21.1 ± 4.8% and 13.9 ± 2.5%, respectively, and those at 10 weeks were 29.0 ± 2.6% and 15.8 ± 2.0%, respectively.

The mechanism of neuronal cell death resulting from the retrograde degeneration of the facial nerve after axonal injury remains unclear; however, apoptosis is reportedly involved [15, 16]. These mechanisms have been suggested to be related to the loss of various neurotrophic factors such as growth factors and cytokines following axonal injury [17, 18]. bFGF administered within the tympanic cavity can affect the facial nerve via several pathways including the direct pathway; however, it involves a perineural structure known as the nerve tissue barrier, primarily comprising tight junctions between perineural cells. Pathological studies of severe Bell's palsy and Hunt syndrome have shown that inflammation of the perineural membrane causes dysfunction of the nerve tissue barrier, leading to the opening of tight junctions [19–21], which is also observed in freezing-induced injuries [11]. Facial canal dehiscence is found in 55–74% cases in healthy human temporal bone pathology [22, 23]. Since bFGF migrates into the facial neural tube through the bony cleft, its intratympanic administration results in migration and interaction with peripheral nerves; however, further research is required to confirm this hypothesis.

The biological half-life of bFGF is less than 50 minutes [24], which is too short for a single topical administration to be effective, whereas daily intratympanic administration of bFGF is highly challenging. Moreover, solutions administered into the tympanic cavity are drained through the Eustachian tube [25]. Therefore, it is essential to ensure that bFGF remains in the tympanic cavity for extended periods. The osmotic pump enables controlled drug release; however, it necessitates pump implantation and catheter fixation in the tympanic cavity. Consequently, they are rarely used in clinical settings. In contrast, bioabsorbable gelatin hydrogels can form polyion complexes with neurotrophic factors, where the collagenase gradually degrades and releases these factors [26]. This method enables a single dose of sustained-release administration over a 2-week period, reducing the burden on the patient.

Significant differences in neuronal cell death processes were observed between the bFGF and control groups at 4 and 10 weeks, but not at 1 week. Dai et al. reported that facial neuronal cell death increased over time, in rats with severed facial nerves in the pores of the internal acoustic meatus, peaking at 15 days after surgery [27]. Although the animal species differed from that used in our study, the authors' results indicate that facial nerve transection at the pores of the internal acoustic meatus is a more proximal and intense injury than temporal

bone infratemporal facial nerve freezing. Moreover, our freezing-induced facial paralysis model suggests that neuronal cell death rate may peak at 2 weeks or later due to retrograde degeneration.

In the present study, neuronal cell death in the facial nerve was 24.1% at 4 weeks and 29.0% at 10 weeks post-treatment, while Marzo et al. reported that compression of the intratemporal portion of the facial nerve in rats resulted in 10.6% cell loss after 4 weeks and 14.2% cell loss after 8 weeks [7]. This discrepancy could be attributed to the differences in the degree of nerve damage. Generally, retrograde degeneration in the peripheral nerves is more likely to occur under the following conditions: 1) higher levels of impairment, 2) damage sites closer to the nerve cell bodies, and 3) in younger animals [28]. The freezing-induced facial paralysis model used in this study was severe. The ENoG value was 0% at 1 week, and residual paralysis was observed at 10-weeks, indicating inadequate natural recovery. Despite such severe paralysis models, intratympanic bFGF administration was found to promote suppression of neuronal cell death.

Histopathological analysis revealed neuronal cell loss in the facial nerve nucleus as well as cells with densely stained cytoplasm and shrunken cell bodies. Staining changes in the cytoplasm reflect chromatolytic changes, which are morphological changes in the neuron that occur as a reaction to axonal injury [29]. This result supports the idea that the entire neuronal population responds to freezing of the intratemporal facial canal. The irregular shape of the nerve cells observed at 10 weeks might suggest that these cells had escaped neuronal cell death. However, the implications of these morphological changes in nerve function require further evaluation. Nonetheless, the shrinkage of nuclei and cell bodies could indicate changes resulting from severe impairment.

Alongside its positive results, this study had some limitations. First, when using the freezing-induced facial paralysis model, the difference in facial paralysis-causing mechanisms from those that cause Bell's palsy or Ramsey-Hunt syndrome should be considered because they reportedly have similar histological features [11], whereas the most significant damage usually occurs in the labyrinthine region of the facial nerve in viral neuritis. However, the site of damage in this model was different. Second, understanding how the administered bFGF reaches the facial nerve and central neuronal nuclei within the tympanic chamber, and how it acts, requires further evaluation, as these mechanisms were not analyzed in this study. Finally, as in previous studies, accurate assessment of nose and lip movements in facial paralysis scoring was challenging, which may have led to an underestimation of the scores. Therefore, facial paralysis scoring should be conducted by two or more researchers to reduce bias and obtain optimal results.

## Conclusion

The percentage reduction of facial nerve neurons was evaluated using a freeze-induced model of facial paralysis. Neuron counts were reduced by the freezing treatment; however, the rate of reduction was suppressed by intratympanic bFGF administration, suggesting its protective effect on facial neurons. Further studies in humans are warranted before clinical application; however, this treatment could reduce the incidence of poor recovery in patients with severe facial paralysis.

## Supporting information

**S1 Table. Dataset of cell count.**
(PDF)

**S2 Table. Dataset of facial movement score (bFGF group).**
(PDF)

**S3 Table. Dataset of facial movement score (Control group).**
(PDF)

## Acknowledgments

We would like to thank Daizaburo Shimizu of the advanced research support center (ADRES) of Ehime University for assistance with preparing the histological specimens.

## Author Contributions

**Conceptualization:** Shinji Iwata, Hiroyuki Yamada, Naohito Hato.

**Data curation:** Shinji Iwata, Takemichi Tanaka, Takuya Kimura, Tomonori Joko.

**Formal analysis:** Masato Teraoka.

**Investigation:** Takemichi Tanaka.

**Methodology:** Shinji Iwata, Hiroyuki Yamada, Masato Teraoka, Takuya Kimura, Tomonori Joko, Hiroyuki Wakisaka, Naohito Hato.

**Project administration:** Hiroyuki Yamada, Naohito Hato.

**Resources:** Yasuhiko Tabata.

**Writing – original draft:** Shinji Iwata, Hiroyuki Yamada.

**Writing – review & editing:** Hiroyuki Yamada, Naohito Hato.

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
