## [Decision Letter · Decision Letter 0]

8 Dec 2024

PONE-D-24-44418Basic fibroblast growth factor helps protect facial nerve cells in a freeze-induced paralysis modelPLOS ONE

Dear Dr. Iwata,

Thank you for submitting your manuscript to PLOS ONE. After careful consideration, we feel that it has merit but does not fully meet PLOS ONE’s publication criteria as it currently stands. Therefore, we invite you to submit a revised version of the manuscript that addresses the points raised during the review process.

The authors should answer the minor modifications suggested by the reviewers in order to enhance the quality and the clarity of the manuscript. One major point is to reduce the number of self-citation. As it is here, the high number of references from the authors is not acceptable. Please answer clearly this point in your rebuttal letter and list the references that have been replaced.

We look forward to receiving your revised manuscript.

Kind regards,

Christophe Egles, Ph.D.

Academic Editor

PLOS ONE

Journal Requirements:

2. You have indicated that data is available from [email address]. Please can we ask you to provide us with a general contact email address for the data requests, so readers can request access in perpetuity. If a general email is not available please provide a link to a website where readers can obtain access to data.

Reviewers' comments:

Reviewer's Responses to Questions

**Comments to the Author**

1. Is the manuscript technically sound, and do the data support the conclusions?

Reviewer #1: Yes

Reviewer #2: Partly

2. Has the statistical analysis been performed appropriately and rigorously? 

Reviewer #1: Yes

Reviewer #2: I Don't Know

3. Have the authors made all data underlying the findings in their manuscript fully available?

Reviewer #1: Yes

Reviewer #2: Yes

4. Is the manuscript presented in an intelligible fashion and written in standard English?

Reviewer #1: Yes

Reviewer #2: Yes

5. Review Comments to the Author

Reviewer #1: The manuscript addresses a clinically significant topic and proposes a solution. Seeking new, simple, and easily accessible treatment methods using the latest knowlesge and technology is extremely important for the development of medicine and improving the quality of life for patients. The authors have correctly selected the analyses, applied appropriate statistical methods, and presented their results concisely and substantively, discussing them with the current literature. Therefore, I recommend the manuscript for publication.

Reviewer #2: This study investigates the induction of facial nerve paralysis and evaluates the potential use of bFGF as a treatment. While the work is interesting, the Materials and Methods, the Results and the Discussion sections require some clarification. Moreover, the authors refer too much on their past researches. Therefore, they should include more references from other relevant studies in the literature.

Abstract

Lines 27-28: Explain if the saline solution was also impregnated into the gelatin hydrogel. This point is not clear.

Line 29: If I understand correctly, electrophysiological testing was conducted one week postoperatively. This could confuse readers since you mentioned that facial movement scoring, electrophysiological testing, and histological assessment of facial neurons were performed at different time points.

Line 33: Replace "control group" with "treated with saline," as you did not mention yet that the « saline treatment » was the control group.

Materials and Methods

Line 76: To enhance the reporting of animal experiments, you can refer and cite the ARRIVE 2.0 guidelines for animal research reporting.

Here is the reference: "The ARRIVE guidelines 2.0: Updated guidelines for reporting animal research" (DOI: https://doi.org/10.1177/0271678X20943823).

Line 83: Did you administer any anti-inflammatory or pain-relief medication, such as buprenorphine, to minimize animal suffering? Please clarify.

Line 90: Specify the dimensions of the gelatin hydrogel. Is the 125 µg mentioned the weight of gelatin hydrogel? This needs to be clearer.

Line 108: As discussed in the Discussion section, report details about the experts scoring the experiments: number of experts? Were they blinded to the group allocations during scoring?

Line 127: Explain better the allocation of the condition within each time point : how many animal per condition ? Since there are histological time points, in the scoring system is it still the same number of animals that is being compare from week 1 to 10 ?

Line 134: Specify which software was used for cell counting.

Line 142: Adjust the description of Figure 2 so that the text explains images "a" and "b" directly after each letter is mentioned. Explain the figure’s annotations, what do the circles refer to ?

Results

Line 163: Provide more context for the curve in Figure 4A. What does the black dotted line represent on image A?

Line 167: In Figure 3, clarify which curves correspond to which groups. Also, replace "point" on the Y-axis with "score" for clarity. Maybe you could consider adding a table in this figure to sum up the scoring system described in the Materials and Methods section.

Line 170: For Figure 4, specify whether the example shown is from the control or bFGF-treated group.

Line 174: Instead of referencing Figure 2 from the Materials and Methods, you could directly refer to Figure 5, which includes the histological slides.

Line 175: On the histological slides, highlight the three cellular components you are describing for better understanding.

Line 180: In Figure 5, since "normal" refers to the healthy side in the control group, maybe you could state here you are on the damage nerve for the control group and bFGF-treated group.

Discussion

Line 218: Justify your choice of the bFGF concentration. You discuss its biological half-life but should also address whether this concentration is based on prior studies. Will this concentration be applicable in clinics? Would increasing the concentration enhance the effect?

Line 235: Explain why in your study you chose 1, 4, and 10 weeks as time points when you anticipated a peak at 2 weeks.

Line 269: This is an interesting remark and it should also be mentioned in the Materials and Methods section. Specify whether the researcher assessing the outcomes was blinded to the group allocations.

6. PLOS authors have the option to publish the peer review history of their article (what does this mean?). If published, this will include your full peer review and any attached files.

Reviewer #1: No

Reviewer #2: No

---

## [Author Response · Author response to Decision Letter 0]

5 Jan 2025

Response to reviewers

December 21, 2024

Emily Chenette

Editor-in-Chief

PLoS ONE

Dear Editor: 

I wish to re-submit the manuscript titled “Basic fibroblast growth factor helps protect facial nerve cells in a freeze-induced paralysis model.” The manuscript ID is PONE-D-24-44418.

We thank you and the reviewers for your thoughtful suggestions and insights. The manuscript has benefited from these insightful suggestions. I look forward to working with you and the reviewers to move this manuscript closer to publication in PLoS ONE.

The manuscript has been rechecked and the necessary changes have been made in accordance with the reviewers’ suggestions. The responses to all comments have been prepared and given below. 

Thank you for your consideration. I look forward to hearing from you.

Sincerely,

Shinji Iwata

Department of Otolaryngology, Head and Neck Surgery, Ehime University School of Medicine

Shitsukawa, Toon, Ehime, 791-0295, Japan

Phone: +81-89-960-5366

Fax: +81-89-960-5368

E-mail: h447033x@mails.cc.ehime-u.ac.jp

Response to editor

One major point is to reduce the number of self-citation. As it is here, the high number of references from the authors is not acceptable. Please answer clearly this point in your rebuttal letter and list the references that have been replaced.

Thank you for bringing this to our attention. We recognize that self-citation at the rate included in the original draft may not be acceptable. We have deleted a reference to Yamada’s report from the discussion section (line 262).

1.When submitting your revision, we need you to address these additional requirements. Please ensure that your manuscript meets PLOS ONE's style requirements, including those for file naming. The PLOS ONE style templates can be found at

We have confirmed that our paper complies with the style requirements of PLOS ONE, as indicated in the templates.

2. You have indicated that data is available from [email address]. Please can we ask you to provide us with a general contact email address for the data requests, so readers can request access in perpetuity. If a general email is not available please provide a link to a website where readers can obtain access to data.

I apologize for not being able to provide an appropriate response multiple time. I have registered it in Ehime University's repository, so I kindly ask for your confirmation.

Ehime University Library, Academic Information Team

(Library Main Building, 2nd Floor, West)

3 Bunkyo-cho, Matsuyama, Ehime 790-8577, Japan

TEL: +81-89-927-8841 (Extension: 8841, 8842)

Email: libsys@stu.ehime-u.ac.jp

https://ehime-u.repo.nii.ac.jp/records/2002710

We have added the missing DOIs to the reference list and removed papers that could not be located.

Reviewer #1: The manuscript addresses a clinically significant topic and proposes a solution. Seeking new, simple, and easily accessible treatment methods using the latest knowlesge and technology is extremely important for the development of medicine and improving the quality of life for patients. The authors have correctly selected the analyses, applied appropriate statistical methods, and presented their results concisely and substantively, discussing them with the current literature. Therefore, I recommend the manuscript for publication.

Response to Reviewer #1

We sincerely appreciate the time, effort, and expertise that the reviewers have invested in evaluating our manuscript. Thank you for your dedication and generous guidance throughout the review process.

Reviewer #2: This study investigates the induction of facial nerve paralysis and evaluates the potential use of bFGF as a treatment. While the work is interesting, the Materials and Methods, the Results and the Discussion sections require some clarification. Moreover, the authors refer too much on their past researches. Therefore, they should include more references from other relevant studies in the literature.

Response to Reviewer #2

Thank you for reviewing our paper. We recognize that self-citation at the rate included in the original draft may not be acceptable. We have deleted a reference to Yamada’s report from the discussion section (line 262).

We have answered each of your questions below.

Abstract

Lines 27-28: Explain if the saline solution was also impregnated into the gelatin hydrogel. This point is not clear.

We added a note indicating that the gelatin hydrogel was impregnated with some saline solution (lines 28- 29).

Line 29: If I understand correctly, electrophysiological testing was conducted one week postoperatively. This could confuse readers since you mentioned that facial movement scoring, electrophysiological testing, and histological assessment of facial neurons were performed at different time points.

Thank you for bringing to our attention this potentially confusing wording. We have removed the statement "at different postoperative time points" for greater clarity (lines 29-30).

Line 33: Replace "control group" with "treated with saline," as you did not mention yet that the « saline treatment » was the control group.

Thank you for your feedback. We have clarified in the abstract that the saline-treated group was indeed the control group (line 29). We refer to the “control” group thereafter in the manuscript. 

Materials and Methods

Line 76: To enhance the reporting of animal experiments, you can refer and cite the ARRIVE 2.0 guidelines for animal research reporting. Here is the reference: "The ARRIVE guidelines 2.0: Updated guidelines for reporting animal research" (DOI:

https://doi.org/10.1177/0271678X20943823).

Thank you for your suggestion. This reference has been added to the reference list.

Line 83: Did you administer any anti-inflammatory or pain-relief medication, such as buprenorphine, to minimize animal suffering? Please clarify.

Ketamine and xylazine were the only anesthetics used in this experiment. The combination of these two agents provided both sedation and analgesia. Before starting the procedure, we used the absence of the pharyngeal reflex as an indicator of suitable depth of anesthesia, which we believe to minimize animal distress (lines 83-86).

Line 90: Specify the dimensions of the gelatin hydrogel. Is the 125 µg mentioned the weight of gelatin hydrogel? This needs to be clearer.

The gelatin hydrogel was prepared as a 5 × 5 mm sheet. The “125 µg” refers to the weight of bFGF (line 93).

Line 108: As discussed in the Discussion section, report details about the experts scoring the experiments: number of experts? Were they blinded to the group allocations during scoring?

We agree with the suggestion that multiple experts should have scored the experiments and that they should have been blinded to group allocation. In this study, only one observer scored the experiments without being blinded to group allocation; we have added this information to the Methods section (lines 116-118).

Line 127: Explain better the allocation of the condition within each time point : how many animal per condition ? Since there are histological time points, in the scoring system is it still the same number of animals that is being compare from week 1 to 10 ?

Thank you for bringing to our attention the aspects of our methodology that may be confusing to the reader. We have revised the Surgical Procedures section for greater clarity (lines 98-102). Specifically, facial movement evaluation was conducted in the bFGF 10-week group and the control 10-week group (n=7 per group) (line 108). The bFGF 1- and 4-week group and the control 1- and 4-week groups (n=7 per group) did not undergo facial nerve paralysis scoring. 

Line 134: Specify which software was used for cell counting.

Cell counting was performed with ImageJ (lines 142-144).

Line 142: Adjust the description of Figure 2 so that the text explains images "a" and "b" directly after each letter is mentioned. Explain the figure’s annotations, what do the circles refer to ?

The descriptions for (a) and (b) were moved to the beginning of the explanatory text, and additional information about the presented circles and numbers was added (lines 152-156).

Results

Line 163: Provide more context for the curve in Figure 4A. What does the black dotted line represent on image A?

An arrow was added between the black dashed lines to indicate the amplitude (line 184 and Figure 4).

Line 167: In Figure 3, clarify which curves correspond to which groups. Also, replace "point" on the Y-axis with "score" for clarity. Maybe you could consider adding a table in this figure to sum up the scoring system described in the Materials and Methods section.

A description was added to indicate that the red and blue lines represent the bFGF and control groups, respectively. Additionally, "point" was changed to "total score," and a table summarizing the scoring system was included (lines 179-180 and Figure 3).

Line 170: For Figure 4, specify whether the example shown is from the control or bFGF-treated group.

The example shown is from the control group; we have included this information in the figure legend (similar results were observed in the bFGF group) (lines 184-185).

Line 174: Instead of referencing Figure 2 from the Materials and Methods, you could directly refer to Figure 5, which includes the histological slides.

The reference in the text was updated from Figure 2 to Figure 5 (lines 188-189).

Line 175: On the histological slides, highlight the three cellular components you are describing for better understanding.

Arrows and corresponding explanations were added for each cell component (lines 189-192 and Figure 5).

Line 180: In Figure 5, since "normal" refers to the healthy side in the control group, maybe you could state here you are on the damage nerve for the control group and bFGF-treated group.

We have added an explanation for the arrows included in Figure 5. We have also included a note clarifying that this explanation refers to the affected side of the control and bFGF groups (lines 195-201).

Discussion

Line 218: Justify your choice of the bFGF concentration. You discuss its biological half-life but should also address whether this concentration is based on prior studies. Will this concentration be applicable in clinics? Would increasing the concentration enhance the effect?

The bFGF concentration used in this study was based on a previous study, which used it at 100 μg/100 μl. We have included this information in the Materials and Methods section (lines 93-95).

In this experiment, we used Fiblast Spray, containing 250 μg of bFGF per bottle, which was halved, with 125 μg/125 ml administered per animal. When impregnated into some gelatin hydrogel for sustained release administration, the dosage remains within the clinically acceptable range, even when adjusted for human body weight. While increasing the concentration may potentially alter the results, such experiments were not conducted in this study and remain a subject for future investigation.

Line 235: Explain why in your study you chose 1, 4, and 10 weeks as time points when you anticipated a peak at 2 weeks.

The evaluation started with facial movement assessment in the 10-week group, which revealed a difference in scores between the control and bFGF groups; additionally, histological findings revealed differences in cell counts. Based on these findings, the next evaluation was conducted at 4 weeks, when significant differences in the facial nerve paralysis scores were first observed. Differences in cell counts were also observed at 4 weeks. The 1-week time point was selected to align with the timing of the electrophysiological examinations. Consequently, evaluations were conducted at 1, 4, and 10 weeks. As a result, it was anticipated that the peak would occur around 2 to 3 weeks.

Line 269: This is an interesting remark and it should also be mentioned in the Materials and Methods section. Specify whether the researcher assessing the outcomes was blinded to the group allocations.

We have included a clarification in the Materials and Methods (Facial Movement Evaluation) section, stating that the outcome was assessed by one researcher who had not been blinded to group allocation.

---

## [Editor Report · Decision Letter 1]

23 Jan 2025

Basic fibroblast growth factor helps protect facial nerve cells in a freeze-induced paralysis model

PONE-D-24-44418R1

Dear Dr. Iwata,

We’re pleased to inform you that your manuscript has been judged scientifically suitable for publication and will be formally accepted for publication once it meets all outstanding technical requirements.

Kind regards,

Christophe Egles, Ph.D.

Academic Editor

PLOS ONE
---

## [Editor Report · Acceptance letter]

28 Jan 2025

PONE-D-24-44418R1 

PLOS ONE

Dear Dr. Iwata, 

I'm pleased to inform you that your manuscript has been deemed suitable for publication in PLOS ONE. Congratulations! Your manuscript is now being handed over to our production team.

Kind regards, 

on behalf of

Professor Christophe Egles 

Academic Editor

PLOS ONE